# India consists of multiple food systems with scoioeconomic and environmental variations

Tushar Ramchandra Athare[1,2]*, Prajal Pradhan[1], S. R. K. Singh[3], Juergen P. Kropp[1,4,5]

**1** Potsdam Institute for Climate Impact Research (PIK), Member of the Leibniz Association, Potsdam, Germany, **2** ICAR ATARI Pune, Pune, India, **3** ICAR ATARI Jabalpur, Jabalpur, India, **4** Institute for Environmental Science and Geography, University of Potsdam, Potsdam, Germany, **5** Bauhaus Earth gGmbH, Berlin, Germany

* athare@pik-potsdam.de

**Data Availability Statement:** The Household Consumer Expenditure Survey (HCES) 2011–12 data used in this study can be requested from

## Abstract

Agriculture in India accounts for 18% of greenhouse gas (GHG) emissions and uses significant land and water. Various socioeconomic factors and food subsidies influence diets in India. Indian food systems face the challenge of sustainably nourishing the 1.3 billion population. However, existing studies focus on a few food system components, and holistic analysis is still missing. We identify Indian food systems covering six food system components: food consumption, production, processing, policy, environmental footprints, and socioeconomic factors from the latest Indian household consumer expenditure survey. We identify 10 Indian food systems using k-means cluster analysis on 15 food system indicators belonging to the six components. Based on the major source of calorie intake, we classify the ten food systems into production-based (3), subsidy-based (3), and market-based (4) food systems. Home-produced and subsidized food contribute up to 2000 kcal/consumer unit (CU)/day and 1651 kcal/CU/day, respectively, in these food systems. The calorie intake of 2158 to 3530 kcal/CU/day in the food systems reveals issues of malnutrition in India. Environmental footprints are commensurate with calorie intake in the food systems. Embodied GHG, land footprint, and water footprint estimates range from 1.30 to 2.19 kg $CO_2$eq/CU/day, 3.89 to 6.04 $m^2$/CU/day, and 2.02 to 3.16 $m^3$/CU/day, respectively. Our study provides a holistic understanding of Indian food systems for targeted nutritional interventions on household malnutrition in India while also protecting planetary health.

## Introduction

Food systems gather all the elements (i.e., environment, people, inputs, processes, infrastructures, and institutions) and activities related to the production, processing, distribution, preparation, and consumption of food, and the output of these activities, including socioeconomic and environmental outcomes [1]. Food systems are currently broken as they fail to provide desirable outcomes, i.e., to end malnourishment, and exhibit no or minimum environmental impacts [2]. In 2019, 8.8% of the population was undernourished, while 39% of the population

http://icssrdataservice.in/datarepository/index.php/catalog.

**Funding:** T. R. Athare's PhD is funded by Indian Council of Agricultural Research, Netaji Subhash ICAR IF 2016-17. P. Pradhan acknowledges funding from Bundesministerium für Bildung und Forschung for the BIOCLIMAPATHS project (Award No 01LS1906A). The funders had no role in study design, data analysis, and preparation of the manuscript.

**Competing interests:** The authors declare no known competing financial interests or personal relationships that could have influenced the work reported in this paper.

are suffering from overweight and obesity globally [3, 4]. Similarly, food systems are responsible for 21–37% of the total global anthropogenic greenhouse gas (GHG) emissions [5, 6].

Looking at India, the world's third-largest anthropogenic GHG emitter [7], the agriculture sector contributes to 18% of its total GHG emissions [8]. This sector is a major user of natural resources of the country, e.g., land [9] and water [10]. At the same time, India is also facing the triple burden of malnourishment [11, 12]. Currently, 14% of its population is undernourished, [3], and 19.7% of its adults suffer from overweight and obesity [13]. Increased overweight population and associated non-communicable diseases have become a public health issue that is widely spread across urban and rural areas [12]. Mainly, overweight and obesity have increased more in women living in rural areas and urban slums compared to non-slum urban areas [11]. Additionally, India has a high share of the undernourished population, mainly among adolescent girls, pregnant and lactating women, and children [14]. Nevertheless, undernutrition has been rapidly reduced in the country over the last decade [11, 14]. Malnutrition is associated more with food quality than its quantity [12]. Thus, there is a need to understand variation in food systems across India for addressing all forms of malnourishment.

Existing studies mostly investigate food systems of Indian households based on a few components (i.e., elements, activities, or outcomes of food systems), mainly on their inter-linkages. For example, income and education are positively associated with calorie, protein, and fat intake, whereas some rural households consume more calories than urban ones in India [15–19]. Similarly, farming families with a large landholding have mostly higher calorie intake than smallholders and households with other occupations [16, 18, 20]. In general, Indian diets have become diverse with an increase in income [17, 21]. Regarding food processing, urban households consume more ready to eat and processed foods than rural ones [19, 22]. The diets' environmental footprints are higher in high-income Indian households and northern India because of a larger amount of dairy products consumption [19, 23, 24]. Although existing studies report various aspects of Indian food systems, a holistic understanding covering all the components is still missing. This understanding is needed to fix the broken food systems.

Our study aims to address the research gaps mentioned above based on three objectives: i) to identify food systems of Indian households considering their different components, ii) to understand socioeconomic factors behind the food systems of Indian households, and iii) to estimate environmental implications of the food systems of Indian households. We obtain these objectives by using openly accessible household data for India. Section 2 provides a detailed description of the data used and the method applied. We present the identified food systems of Indian households with their socioeconomic factors and environmental implications in Section 3. Our findings are discussed in Section 4, with our conclusion in Section 5.

## Materials and methods

We use the latest 68[th] round of Household Consumer Expenditure Survey (HCES) 2011–12 for our food system analysis. This dataset provides a contemporary understanding of the food systems of Indian households based on various components. It is India's latest openly available household survey data for food consumption. The survey has information on the consumption of different food items, including fish, dairy, egg and meat (See S2 Table in S1 Appendix). It also includes information on whether food is home-produced, subsidized, or purchased. Various socio-economic features of the household are also available in the survey dataset [25]. These features include location, rural-urban sector, education, occupation, income, food expenditure, household size, and landholding, The HCES administered Type 1 and Type 2 schedules, varying in the reference period for food intake to different households [26]. Schedule Type 1 has a reference period of the last 30 days for all food items. Schedule Type 2 has a

reference period of the previous 30 days for some food items (e.g., cereals, pulses, dairy products, and sugar) and the last seven days for the remaining ones (e.g., fruits, vegetables, spices, eggs, fish, meat, and beverages). We use data from both schedules for our study. The HCES provides information of 101,662 Type 1 and 101,651 Type 2 households (i.e., 203,313 households in total) from 7,469 villages and 5,268 urban blocks.

## Food system indicators

We identify the food systems of Indian households based on 15 indicators from six food system components: consumption, production, policy, processing, environmental footprints, and socioeconomic factors. Additionally, we use four socioeconomic factors to explain and interpret the identified food systems of Indian households.

**Food consumption.** We account for household food consumption by considering intake of total calories, calories from market purchased food, total proteins, animal proteins, total fats, and dietary diversity. The calorie, protein, and fat intake denote the nourishment status of Indian households. We use the nutrient content data from [26] to estimate calorie, protein, and fat intake based on the amount of food consumption. The HCES provides information on food consumption at the household level. The household has members of different ages and sex with varied energy requirements. We convert household members into the consumer unit (CU) (See S1 Table in S1 Appendix), accounting for food distribution within the household [26]. The HCES attributes meals served to guests, employees, and non-household members to serving household consumption. This method inflates the consumption of donor households whereas reducing recipient household consumption. To address this anomaly, we adjust households' food consumption by applying the formula provided by the NSSO [26] (See S1 Equation in S1 Appendix).

Dietary diversity represents the quality of household food intake [27]. We estimate dietary diversity by applying Simpson's index of dietary diversity (SIDD) at a household level based on the 12 food groups [28] (See S2 Table in S1 Appendix). These food groups are cereals, roots, tubers, vegetables, fruits, meat including poultry and offal, eggs, fish and seafood, pulses including legumes and nuts, milk and milk products, oil and fats, sugar including honey, and miscellaneous. SIDD accounts for the share of various food groups in the total calorie intake of the household. This index has been used previously to measure the dietary diversity of households [17, 21, 29]. SIDD value ranges from 0 to 1, with zero indicating all calories from one food group. A score of one denotes an equal contribution of all food groups to the total calorie intake (see Eq 1, where $S_C^2$ denotes squares of shares of food groups to the calorie intake).

$$SIDD = (1 - \sum S_c^2) \qquad (1)$$

**Food production.** Food production also affects the nutrition status of those who produce food for their consumption [30]. Additionally, farm production diversity is positively associated with the dietary diversity of the farming households [31–35]. Household food production is an important component of Indian food systems as India produces most of its food locally [36, 37]. Therefore, we also use calorie consumption from home-produced food, land cultivated, and food production diversity as indicators to identify the food systems of Indian households. Here we consider land cultivated as an indicator instead of land owned to account for India's agricultural purposes of land leasing [38].

Our food production diversity is based on the same 12 food groups used to calculate dietary diversity [28]. We estimate the Simpsons Index of food production diversity (SIFPD) of the households using the Eq 2. Here, $S_h^2$ denotes the squares of calorie shares of home-produced

food groups to total home-produced calories. This index has been used in previous studies [31, 32]. SIFPD accounts for food groups grown and the relative share of a food group to the total home-produced calories.

$$SIFPD = (1 - \sum S_h^2) \tag{2}$$

**Food policy.** The government of India provides subsidized food to the population below the poverty line in India, mainly rice, wheat, and sugar [39]. We estimate the calories received by households from subsidized food to cover the Indian food systems' food policy component.

**Food processing.** We consider the consumption of processed and ready to eat food by households as an indicator of food processing. Diets based on processed and ready to eat food are prevalent in India's urban households [19]. This indicator consists of calories from ready to eat food items, e.g., snacks, beverages, processed products, and meals consumed outside the home and at the workplace (See S3 Table in S1 Appendix).

**Environmental footprints.** We use the GHG emissions, land footprint, and water footprint values for Indian households from [19]. [19] has adapted environmental intensity from [24]. The data on the GHG emissions and land and water footprints are available at the household level according to the HCES's household IDs. However [19], has not estimated the environmental footprints of ready to eat and processed food items due to the unavailability of GHGs, land, and water intensity.

**Socioeconomic factors.** We use the household's daily per capita expenditure as a proxy for their income. The share of total household expenditure spent on food is used to assess food affordability. Food affordability reflects the relative cost of food compared with a household's income and purchasing power [30]. Since food consumption is highly influenced by income [4], we consider this share to identify the food systems of Indian households. Further, our study uses four other socioeconomic factors to explain and interpret the identified food system. These factors, namely education, occupation, rural-urban sector, and location, affect various food system components like food consumption and access to processed foods [18, 19].

## Data processing

The HCES data consists of some extreme values, e.g., unrealistically very high- or low-calorie intake [19]. Many analyses, e.g., k-mean clustering, are sensitive to such extreme values and may affect the results. For this, we truncate outliers for calorie intake, the share of food expenditure, and cultivated land indicators of the data. For calorie intake, we retain the households from 5th to 95th percentiles with 1688 to 4220 kcal/CU/day as followed by [19]. We also discard data of the lowest and highest one percentile households on the share of food expenditure variable. This criterion results in households with a share of food expenditure from 18 to 78% for further analysis. The average operational landholding in India is 1.08 hectares [38]. For the cultivated land variable, we exclude households with twice the landholding of large farmers, i.e., 20 hectares [38]. After removing household data fitting in any of the above exclusion criteria, we use 179479 households for our analysis.

## Cluster analysis

We apply k-means cluster analysis to the household data with 15 indicators to identify Indian food systems. The k-means analysis derives clusters based on the mean value and groups individuals into mutually exclusive clusters [40]. We use the silhouette method for determining the optimal number of clusters from the data. This method measures the similarity of an object with its cluster against the other clusters [41]. The silhouette values vary from -1 to +1, with a

high value indicating the household's assignment to the right cluster. We select 10 clusters (i.e., Indian food systems) as evident from the highest average silhouette value (See S1 Fig in S1 Appendix).

To understand households' nourishment status across India, we compare the identified food systems' calorie, protein, and fat intake against India's recommended intake. The recommended calorie intake for sedentary, moderate, and heavy work in India is 2320, 2730, and 3490 kcal/capita/day, respectively [42]. For a healthy diet, the World Health Organisation recommends that 15–30% and 10–15% of dietary energy should come from fats and proteins, respectively [43]. That means the recommended values for fat intake for sedentary, moderate, and heavy work in India are 39–77 g/capita/day, 46–91 g/capita/day, and 58–116 g/capita/day, respectively. Similarly, the recommendation for protein intake is 58–87 g/capita/day, 68–102 g/capita/day, and 87–131 g/capita/day for sedentary, moderate, and heavy work, respectively.

Further, our study explains and interprets the identified food systems with the four socioeconomic factors, mainly by investigating their food systems distribution. We use the education classes: non-literate, literate, up to higher secondary school, diploma, graduate, and postgraduate and above. The occupation classes for the rural sector are: self-employed in agriculture, self-employed in non-agriculture, regular wage or salary, casual labour in agriculture, casual labour in non-agriculture, and others. The urban sector occupation classes are self-employed, regular wage or salary earning, and casual labour. We also estimate the share of rural and urban households across the identified food systems. Finally, our study investigates Indian food systems' spatial distribution by mapping their shares at the district level.

## Results

We identify ten food systems of Indian households by applying the k-means cluster analysis (See S1 Fig in S1 Appendix). The main source of calories varies across these food systems (See S2 Table in S1 Appendix). Therefore, we classify the ten food systems of Indian households into three categories: production-, subsidy-, and market-based food systems, accounting for their shares of calories.

### Production-based food systems

We consider the three food systems $P_A$, $P_B$, and $P_C$ production-based because home-produced foods contribute to at least 40% (Fig 2). Among these food systems, $P_A$ and $P_B$ have the highest share of subsidized and purchased food, i.e., around 30% and 50% of the calorie intake, respectively. Spatially, these food systems are prevalent in various parts of India (see Fig 1). The food system $P_A$ is present sporadically across Karnataka, Chhattisgarh, Odisha, Jammu and Kashmir, Himachal Pradesh, Uttarakhand, and north-eastern states of India. Food system $P_B$ is prevalent to a varying degree in most of India except Kerala, Tamil Nadu, Andhra Pradesh, and most north-eastern states. Food system $P_C$ is predominant across northern, central, and eastern India and Arunachal Pradesh, Nagaland, Manipur, and Assam.

The total calorie intake in these food systems is larger than the recommended value for sedentary work (i.e., in case of $P_A$) and moderate work (i.e., in case of $P_B$ and $P_C$). The majority of the households ($\approx$45–60%) in the production-based food systems are involved in farming occupation (Fig 3), which is moderate to heavy work [44]. Hence, we consider households with the food system $P_A$ with lower than recommended calorie intake for moderate work as undernourished. In contrast, the food systems $P_B$ and $P_C$ have enough calorie intake according to the occupation. These food systems' protein and fat intake are within the lower end of the recommended range for moderate work. Processed and ready-to-eat food contributes less

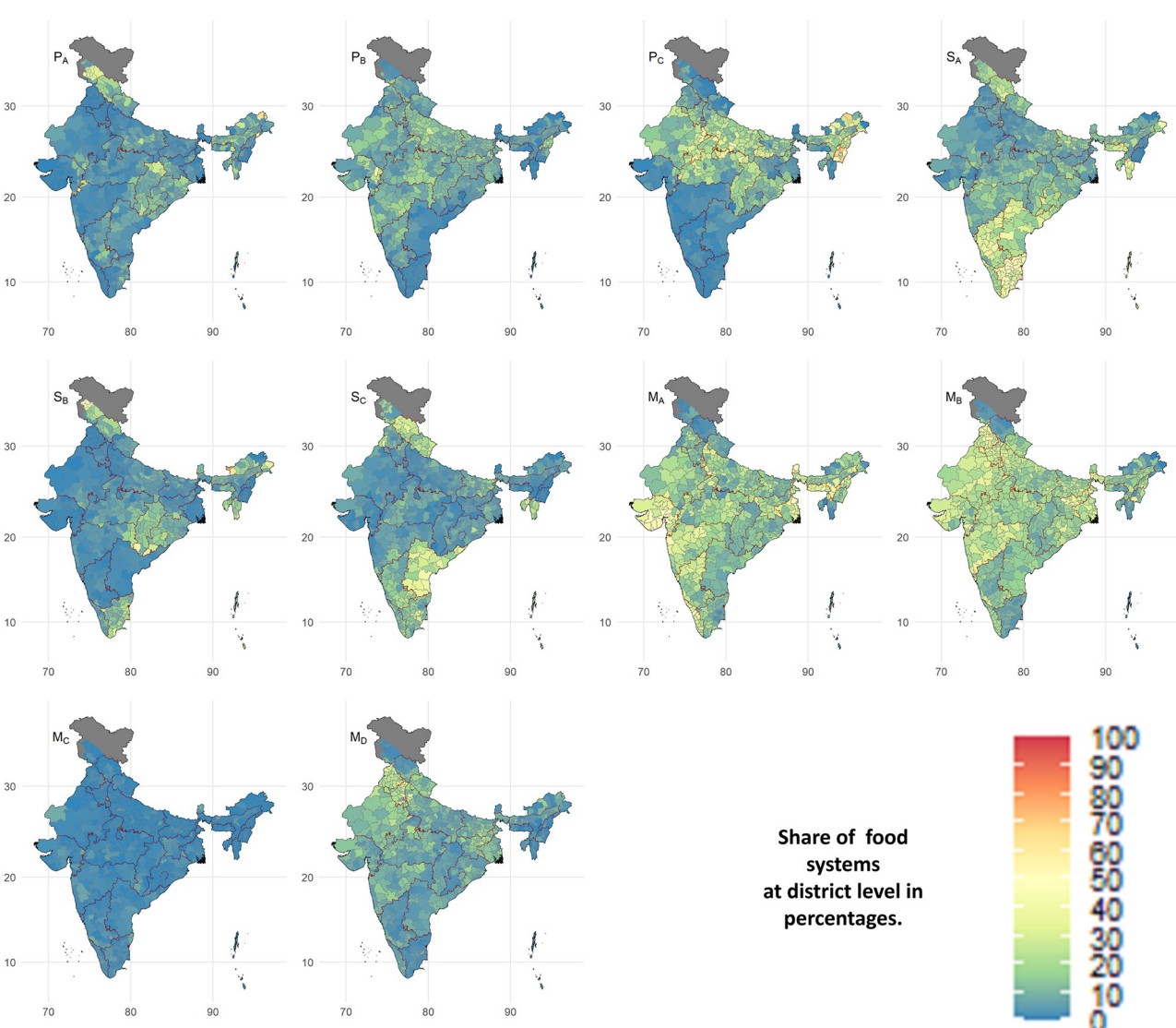

**Fig 1. Spatial distribution of Indian food systems.** The figure describes the percentage share of 10 food systems at the district level. Two states (Andhra Pradesh and Jammu and Kashmir), bifurcated after 2011–12, are considered original states. The data for three districts of Jammu and Kashmir state, i.e., Leh(Ladakh), Kargil, and Poonch, were not available.

than 5% of the total calorie intake in these food systems. This result highlights the lower reliance on processed and ready to eat food among farming families.

Share of calories from production, subsidy and market affects households' dietary diversity (Fig 2). SIDD varies from 0.58 to 0.67 in the production-based food systems. The food system $P_B$ consumes a more diverse diet than the other two because it has the largest share of calories from the market. Market-purchased foods contribute 743, 1498, and 769 kcal/CU/day in the food system $P_A$, $P_B$, and $P_C$, respectively. Although the food system $P_A$ produces the most diverse food with a SIFPD of 0.28, its dietary diversity is the lowest. SIFPD is 0.23 in food system $P_B$ and 0.26 in food system $P_C$. Because of these low values, SIFPD did not translate into SIDD in these food systems. In order words, the current food production diversity of the farmers is not diverse enough to enrich their dietary diversity, either due to lower production

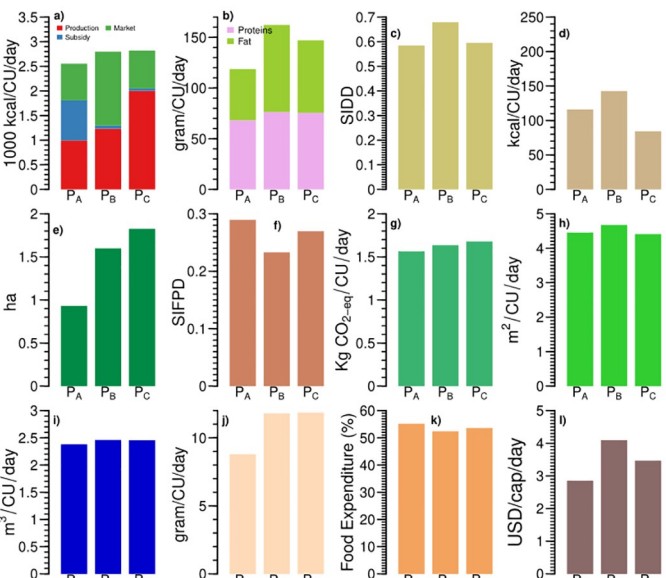

**Fig 2.** Three production-based food systems ($P_A$, $P_B$, and $P_C$) of India vary across different food systems' indicators: (a) sources of calorie intake, (b) protein and fat intake; (c) Simpsons Index of Dietary Diversity (SIDD); (d) processed and ready to eat food intake, (e) land cultivated (ha), (f) Simpsons Index of Food Production Diversity (SIFPD), (g) greenhouse gas emissions, (h) land footprint, (i) water footprint, (j) animal protein intake, (k) share of food expenditure, and (l) daily Per capita income.

amounts or growing only a few food groups. Lower dietary diversity among these households could be addressed through interventions like kitchen gardening.

Within these food systems, the total calorie intake and home-produced calories increase with increased cultivated land from 0.93 to 1.82 hectares. The food system $P_A$ consists of smallholders with less than one hectare of cultivated land (Fig 2). Around 80% of the Indian farmers are smallholders with a landholding below one-hectare [38]. The three production-based food systems represent around 21% of the Indian households (See S4 Table in S1 Appendix). Overall, 570.28 million of the Indian population live in 118.80 million farming households [45].

Environmental footprints vary across the production-based food systems (Fig 2). Their embodied GHG emissions are comparable with their calorie intake, with a range of 1.56–1.67 kg $CO_2$eq/CU/day. However, water and land footprints are not proportional to calorie intake. The food system $P_B$ with lower calorie intake has higher water and land footprints than the food system $P_C$. This variation may be due to the difference in the composition of diets. Since calorie, fat, and protein intake in all three production-based food systems are within or lower than the recommended values, sustainable agricultural production would be a response option for these food systems to lower their environmental footprints.

The three food systems also differ across socioeconomic variables (Fig 3). The food system $P_A$ has the lowest income of 2.85 USD/capita/day and spends the highest share (55%) of its income on food among the ten food systems. Overall the production-based food systems spend more than half of their income on food. At the same time, the share decreases as income increases. The majority ($\approx$45–60%) of the households with production-based food systems are self-employed in agriculture and reside in rural areas. It reflects the dependence for livelihood on the agriculture sector in rural India and the low-income potential of this occupation. Education of up to higher secondary predominates among males and females of the food systems $P_A$, $P_B$, and $P_C$. Overall, the male household members are more educated, especially beyond the diploma and above educational levels.

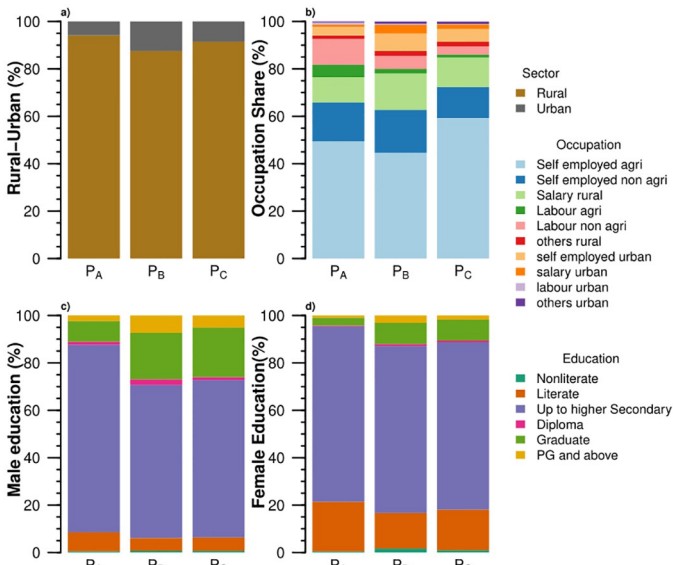

**Fig 3.** Socioeconomic features of Production-based food systems ($P_A$, $P_B$, and $P_C$) of India: (a) the share of rural-urban households, (b) the share of occupation classes, (c) the share of the highest education among male members, and (d) the share of the highest education among female members.

## Subsidy based food systems

We classify the food systems $S_A$, $S_B$, and $S_C$ into subsidy-based food systems due to a large contribution (i.e., 26–62%) of subsidized food in their total calorie intake (Fig 4). The purchased foods provide most of the remaining calories with a meagre share from the home-produced foods. The subsidy-based food systems are predominant mainly in India's southern and eastern regions (see Fig 1). Additional food subsidies by the state government over and above India's government lead to the prevalence of subsidy-based food systems in these regions [46]. The food system $S_A$ is prevalent in Kerala, Tamil Nadu, Karnataka, Andhra Pradesh, Odisha, Himachal Pradesh, Jammu and Kashmir, Mizoram, Tripura, and Meghalaya. The food system $S_B$ prevails in Chhattisgarh, parts of Tamil Nadu and Odisha, and Maharashtra's eastern districts. The states of Andhra Pradesh, Uttarakhand, Mizoram, Tripura, and Himachal Pradesh consist of the food system $S_C$.

The food systems $S_A$ and $S_B$ have lower than the recommended value of calories for sedentary and moderate work. However, the food system $S_C$ has higher calories than recommended value for moderate work. The calorie intake is according to the recommendation for the households engaged in self-employment and labour in agriculture (around 15%) in the food system $S_C$ (Fig 5). However, households engaged in self-employment and agricultural labour in the food system $S_B$ are undernourished, with lower than recommended calorie intake for moderate work. These results highlight that several households with a large share of subsidized foods are malnourished. They are either suffering from undernourishment or overconsumption. Interestingly, protein and fat intake in these food systems are within the range of sedentary work recommendations. Animal-source provides 7.77–11.33 g/CU/day of proteins. We see a low processed and ready to eat food consumption among these food systems (3–6% of calories).

In the subsidy-based food systems, dietary diversity decreases as the share of subsidized food increases, with SIDD varying from 0.56 to 0.64 (Fig 4). The dominance of cereals in

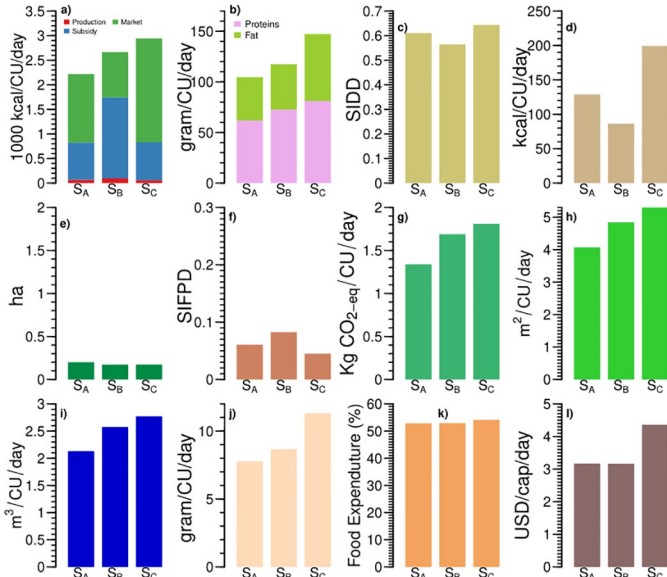

**Fig 4.** Three subsidy-based food systems ($S_A$, $S_B$, and $S_C$) of India differ across food systems' indicators: (a) sources of calorie intake, (b) protein and fat intake; (c) Simpsons Index of Dietary Diversity (SIDD), (d) processed and ready to eat food intake, (e) land cultivated (ha), (f) Simpsons Index of Food Production Diversity (SIFPD), (g) greenhouse gas emissions, (h) land footprint, (i) water footprint, (j) animal protein intake, (k) share of food expenditure, and (l) daily per capita income.

subsidized foods leads to low dietary diversity in these food systems. For example, the food system $S_B$ with a higher subsidized calorie intake has lower dietary diversity than the food system $S_A$. However, a higher share of calories from market-purchased food increases dietary diversity. These food systems represent 30.23% of Indian households (See S4 Table in S1 Appendix). Because of a low share of home-produced foods, these food systems also have a low SIFPD, i.e., below 0.1.

Looking at subsidy-based food systems' environmental footprint, they are commensurate with the calorie intake and vary widely (Fig 4). Embodied GHG emissions in subsidy based food systems are 1.33 to 1.81 kg $CO_2$eq/CU/day. Water footprints of these food systems range from 2.13 to 2.77 m$^3$/CU/day, whereas land footprint is 4.06–5.29 m$^2$/CU/day. The wide variation among the environmental footprints is due to the coexistence of undernourishment and overconsumption in these food systems. Demand-side measures like reducing overconsumption can lower the environmental footprints of these food systems. Better targeting food subsidies could help reduce overconsumption in the food system $S_C$.

We observe similarities and differences among the subsidy-based food systems regarding associated socioeconomic factors (Figs 4 & 5). The food systems $S_A$ and $S_B$ have identical incomes and shares of food expenditure. However, calorie intake in the food system $S_B$ is higher than $S_A$ due to higher calories from subsidized food. The subsidy-based food systems spend 52.86–54.15% of their income on food. The subsidy-based food system $S_C$ has the highest income and the highest share of food expenditure, mainly due to more calories from market-purchased food. These food systems are predominantly rural, with self-employment and labour in non-agricultural activities as the principal household occupations. Urban households with these food systems are mostly self-employed, followed by salaried jobs and labour activities. Education up to higher secondary level is most common among both male and female

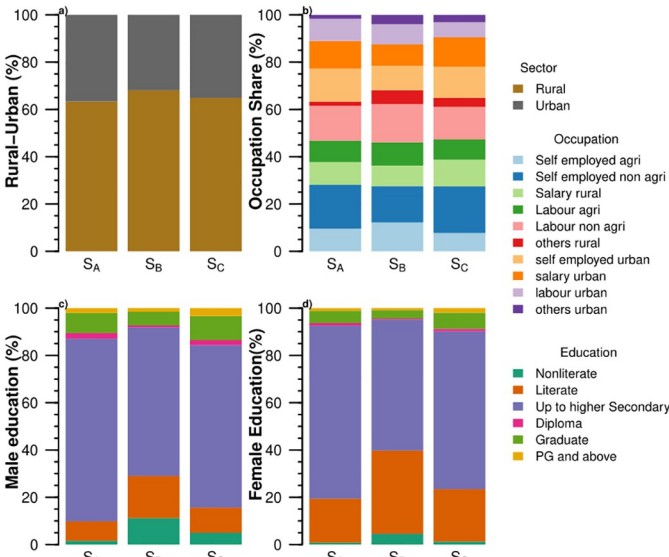

**Fig 5.** Socioeconomic features of Subsidy-based food systems ($S_A$, $S_B$, and $S_C$) of India: (a) the share of rural-urban households, (b) the share of occupation classes, (c) the share of the highest education among male members, and (d) the share of the highest education among female members.

members of the households. Overall, we see higher education among male members than the female.

## Market-based food systems

The four food systems ($M_A$, $M_B$, $M_C$, and $M_D$), depending mostly on purchased food, are market-based (Fig 6). Looking at the spatial distribution, the food system $M_C$ is spread thinly and uniformly across India (see Fig 1). Food system $M_A$ is prevalent in Kerala, Maharashtra, Gujarat, Meghalaya, Sikkim, and partly in Karnataka, Jharkhand, Assam West Bengal. The food system $M_B$ is prevailing across Maharashtra, Madhya Pradesh, Rajasthan, Punjab, Jharkhand, Nagaland, and Manipur. Food system $M_D$ is observed mainly in Punjab, Haryana, western Uttar Pradesh, and sporadically in Arunachal Pradesh, Madhya Pradesh, Maharashtra, Jharkhand.

The food system $M_A$ has lower than recommended calorie intake for sedentary work, reflecting undernourishment among the households. The food systems $M_B$, $M_C$, and $M_D$, exceed the recommended values for a calorie intake of moderate work resulting in overconsumption. Households with the food systems $M_B$, $M_C$ and $M_D$ are involved in sedentary to moderate work like self-employment, salaried jobs, and non-agricultural activities (Fig 7). The overconsumption in India is also reflected by the prevalence of overweight among 19.7% of adults [47].

Fat intake in these food systems is high. Similarly, protein intake also exceeds the outer bound of recommendation for sedentary work. Protein and fat intake in the food system $M_A$ is according to the lower end recommendation for sedentary and moderate work. The food systems $M_B$, $M_C$ and $M_D$ have protein intake according to moderate, sedentary, and heavy work. Animal-source provides 8.78 to 17.45 g/CU/day of proteins in these food systems.

The households in the food system $M_C$ depend on processed and ready to eat foods for 93% of the calories and live in the urban sector (Figs 6 and 7). The other three food systems, $M_A$,

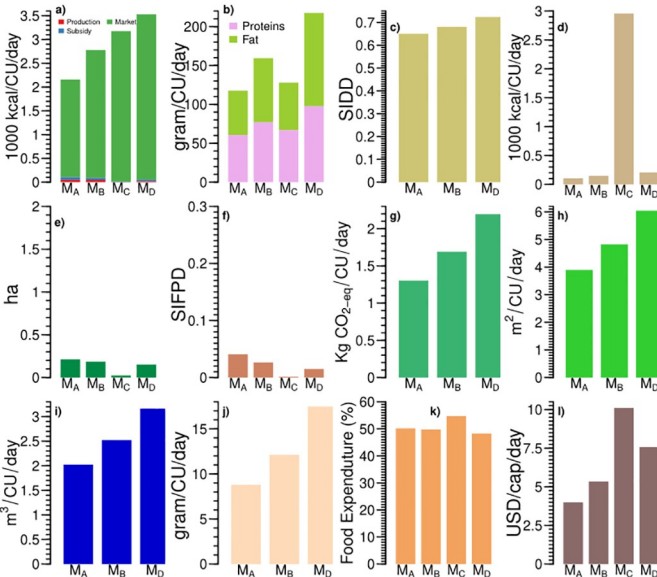

**Fig 6.** We see a variation across food systems indicators among four market-based food systems ($M_A$, $M_B$, $M_C$, and $M_D$) of India: (a) sources of calorie intake, (b) protein and fat intake; (c) Simpsons Index of Dietary Diversity (SIDD), (d) processed and ready to eat food intake, (e) land cultivated (ha), (f) Simpsons Index of Food Production Diversity (SIFPD), (g) greenhouse gas emissions, (h) land footprint, (i) water footprint, (j) animal protein intake, (k) share of food expenditure, and (l) daily per capita income.

$M_B$, and $M_D$, have 5% calories from processed and ready to eat foods. Dietary diversity increases with an increase in income. It varies from 0.64 to 0.72 in the market-based food systems $M_A$, $M_B$, and $M_D$ (Fig 6). Almost half of the Indian households (48%) meet their dietary requirements mainly through market purchased foods. These food systems have a negligible share of home-produced foods with low SIFPD, i.e., below 0.05.

Environmental footprints are proportional to the calorie intake in these food systems with a higher variation. We see the lowest and highest environmental footprints among ten food systems in market-based food systems. These food systems have embedded GHG emissions ranging from 1.30 to 2.19 kg $CO_2$eq/CU/day. Three market-based food systems use 2–3.16 m$^3$/CU/day of water and 3.89 to 6 m$^2$/CU/day of land. Overconsumption among three and undernourishment in one, market-based food systems causes wide variation in environmental footprints. Reducing overconsumption could lower the environmental footprints of these food systems.

Among socio-economic factors, the share of food expenditure decreases with an increase in income, except for food system $M_C$ (Fig 6). The food system $M_C$ with the highest income has the highest share of food expenditure. Relying on processed and ready to eat foods explains a higher share of food expenditure in the food system $M_C$. The market-based food systems are predominantly urban. The main household occupations in these food systems are self-employment and salaried work for the urban sector. Households in the food system $M_C$ are also engaged in other occupational activities in the urban sector. Rural sector households are engaged in non-agricultural activities and salaried jobs. The food system $M_C$ has the highest share of non-literate males and females. More than 80% of the female are non-literate in the food system $M_C$ (Fig 7). Calorie intake increases educational levels within other market-based food systems for males and females.

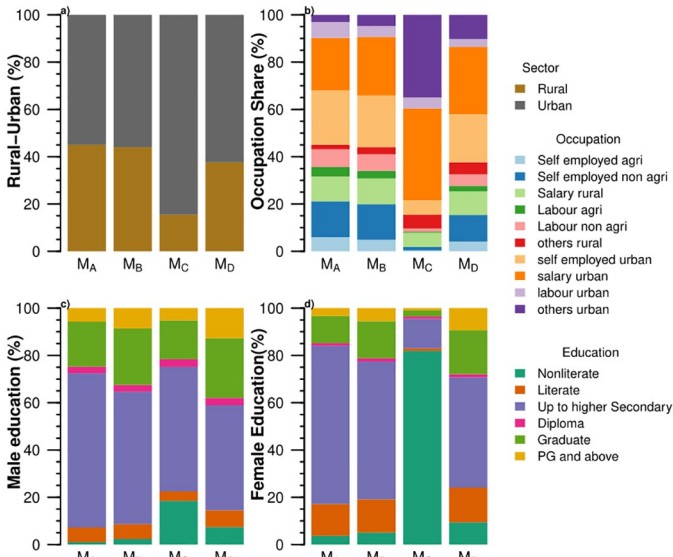

**Fig 7.** Socioeconomic features of Market-based food systems ($M_A$, $M_B$, $M_C$ and $M_D$) of India: (a) the share of rural-urban households, (b) the share of occupation classes, (c) the share of the highest education among male members, and (d) the share of the highest education among female members.

## Comparison of Indian food systems

India faces malnutrition with a prevalence of undernourishment and overconsumption among the food systems. Our study finds undernourishment in the production-, subsidy- and market-based food systems of India. However, overconsumption is prevalent among subsidy- and market-based food systems. Protein and fat intake is within the recommendation for production- and subsidy-based food systems. However, overall protein and fat intake in market-based food systems are higher than recommended for moderate work. Market-based food systems have a higher dietary diversity. Among production- and subsidy-based food systems, dietary diversity increases with increased market calories. Processed and ready to eat food consumption is lower in food systems with a higher share of home-produced and subsidized food. The majority of the household with production-based food systems are engaged in agriculture. The subsidy- and market-based food systems households are employed mostly in non-agricultural occupations.

Indian food systems vary widely in environmental footprints, especially among market-based and subsidy-based food systems. These variations mainly result from a large difference in calorie intake within these food systems. Malnutrition co-exists within these food systems with undernourishment (food systems $S_A$ and $M_A$) and overconsumption (food systems $S_C$, $M_B$ and $M_D$). Overall GHG emissions are proportionate to the calorie intake in Indian food systems. However, for water footprints, it is not the case. The composition of diets with variation in amounts of environmental footprint intensive products, e.g., rice and animal-sourced foods, are responsible for varying environmental footprints [19]. A multi-faceted approach to addressing malnutrition, dietary changes, and sustainable food production may help make Indian food systems more sustainable.

Households across the food systems spend around half of their income on food. Generally, the share of food expenditure decreases with an increase in income. However, the share of expenditure on food is nearly similar in the food systems $P_A$ and $M_C$. Although the food

system $P_A$ has the lowest income, in contrast to the highest income of the food system $M_C$. Processed and ready eat foods consumption is responsible for a higher share of food expenditure in the food system $M_C$. Production- and subsidy based food systems are predominantly rural, and market-based food systems are dominated the urban sector. Female members are less educated than their male counterparts in all the food systems. These food systems are spread to varying degrees across India. Although, the subsidy-based food systems are distinctive in terms of their predominance in the southern states of India.

## Discussion

Our study presents a novel analysis identifying ten food systems of Indian households covering food production, consumption, processed food consumption, food policy, environment, and socioeconomics components. Home-production, subsidized, and market-purchased food are three sources of calorie intake in India. We group the ten food systems into the production-, subsidy-, and market-based food systems. The food systems analysis reveals malnutrition in India, with overconsumption resulting in higher environmental footprints. We highlight household food sources' underlying roles like production for home consumption, subsidy, and market-purchase on their nourishment in India.

Our results reveal undernourishment among all three Indian food systems. In contrast, overconsumption is found only in subsidy- and market-based food systems. Three food systems, $P_A$, $S_A$, and $M_A$, have lower than recommended calorie intake for their occupation. Many Indian households are undernourished, although they are receiving subsidized foods from the government [48].

Overconsumption in India is mainly related to market-purchased food, as seen in three market-based and one subsidy-based food system. Overall the households with these food systems have a higher income. Overconsumption among high-income households in India has been reported previously [17, 18]. We also find undernourishment among households with higher incomes, e.g., in the food system $M_A$. Existing studies also report undernourishment among some of the high-income households in India [15, 18, 19]. Low awareness about healthy diets and sedentary lifestyles could be the reasons for undernourishment among these households.

We report higher than recommended protein intake for a lower bound of sedentary work in all Indian food systems. These results contrast with lower protein intake among low-income Indian households in previous studies [15, 17]. The difference might be due to using data from Type 1 and Type 2 schedules to identify the food systems. Schedule type 2 has higher protein and fat intake values due to a lower recall period of 7 days for some food items. However, the results of higher than lower bound of recommended fat intake for moderate work in most of the Indian food systems confirms previous study findings [17].

Our study reveals that food sources affect household dietary diversity. Home-production and subsidized food calories are negatively associated with the dietary diversity of Indian households. Existing studies only focus on the role of income on dietary diversity in India [17, 21].

For the first time, we analyze the home-produced foods' role in Indian food systems via contribution to the calories, diversity of food production, and the cultivated land variables. We found that calorie intake increases with a high share of home-produced calories and landholding. However, a high production diversity did not translate into a high dietary diversity. These food systems have an overall low production diversity with SIFPD value below 0.3. Thus, adequate strategies are needed to increase production diversity, e.g., kitchen gardening, to improve the dietary diversity of Indian households largely dependent on home-produced

food. Currently, market purchased food contributes to higher dietary diversity among farm families in India. An earlier study reported higher calorie intake with an increase in the land-holding in India [20]. Existing studies report enhanced dietary diversity with market access in farming households for other countries [33–35].

Our study report that environmental footprints, i.e., embodied GHG emissions and land and water footprints, of Indian food systems are proportionate to the calorie intake. Our estimates of GHG emissions of Indian food systems are comparable to most Indian diets [19, 24]. However, few Indian diets have higher GHG emissions than our study [24]. This difference is mainly due to varying dietary composition [24]. Our reported land footprint for Indian food systems is slightly higher than earlier studies' land footprint [19, 24]. The varying composition of the diets could lead to different land footprints. Globally the land footprints of Indian food systems are lower than the United States of America diets [49] and higher than the Chinese diets [50]. Our study finds a higher environmental footprint in the north Indian states of Punjab, Haryana, and Western Uttar Pradesh. These regions of India have overconsumption, with higher dairy products consumption, resulting in higher environmental footprints [19, 23, 24]. Dairy product consumption needs to be reduced in these regions. Reduced dairy products consumption should be supplemented with ruminant numbers reduction to avoid the rebound effect [51].

Around 77% of the Indian households are malnourished, with 38% being undernourished and 39% have overconsumption. Adopting healthy diets may slightly increase the environmental footprints in India [24]. Healthy diets and supply-side measures like closing yield gap, sustainable intensification, and judicious input use are necessary to keep food systems within planetary boundaries [5, 6, 19, 52–56]. Sustainable and healthy diets provide climate change mitigation and adaptation synergies, along with health co-benefits [5, 6, 19, 55].

We explore various socioeconomic factors, e.g., rural-urban sector, income, occupation, and education, affecting Indian food systems. Our results of higher undernourishment in rural India are similar to previous studies [17, 18]. We report higher processed and ready to eat food consumption in urban areas, thinly spread all over India. Previous studies also have similar results for processed food consumption in India [19, 22]. Our results of a higher prevalence of subsidy based food systems in the rural sector confirm the earlier study finding [48]. The finding of higher calorie intake among households with higher education in our study is similar to the earlier reported results [18].

Our study also has some limitations. The HCES gives the expenditure of households on food, and it does not provide further information on food intake. Food waste has increased in India in recent decades [57] and may affect actual food intake. However, HCES is the only nationally representative data for food intake in India. Since we used the HCES 2011–12 data, the food systems we identified are around ten years old. Thus, there is a need to use the recent household survey data, once they are available, to understand the current food systems because food systems are changing. Further, analysis of HCES data from different periods would also provide new insights on changes in food systems across India. Additionally, we could not estimate dietary diversity, animal protein intake, and environmental footprints for the food system $M_C$ due to the unavailability of the composition of ready to eat and processed food items. However, the food system $M_C$ represents a 1.63% population; hence we explain these indicators for most of the data. Our analysis limits our explanation of the food systems' environmental footprints because of not considering dietary composition. Investigation of the dietary composition could better explain the reasons behind variation on these footprints [19]. Instead, we infer findings from Athare and colleagues [19] while interpreting our result, which is also our data source.

Our study of Indian food systems could help better target policies according to different food systems as tailored interventions to address malnutrition, dietary diversity, and environmental sustainability issues. Production-based food systems face undernourishment, whereas home-produced food is not contributing to dietary diversity. Policy interventions like kitchen gardening among farm families may help increase dietary diversity, which is currently low. Kitchen gardening could include the production of seasonal vegetables and perennial fruits to meet the nutritional needs of the households. Additionally, these households need support to increase their agricultural productivity and off-farm incomes to supplement home-produced food when it is not enough to nourish throughout the year. Interventions to increase food production need to be carefully designed to have synergistic effects on social, economic, and environmental systems, tackling the current sustainability trade-offs of food systems [58]. Subsidy-based food systems face undernourishment and overconsumption with lower dietary diversity. Here, policies on better-targeting food subsidies, dietary awareness, and diversifying food subsidies from cereals will help address malnutrition and dietary diversity issues. Focused policies on healthy diet awareness in middle-class families would help address overconsumption in market-based food systems. Here, undernourishment needs to be addressed by tackling issues related to urban poverty. Reducing overconsumption and lower animal protein intake will transform Indian food systems into healthy and sustainable ones.

## Conclusion

We identify and classify 10 Indian food systems into production-, subsidy- and market-based food systems. We find malnutrition in India with the coexistence of undernourishment and overconsumption. Three Indian food systems are undernourished, one each in production-, subsidy- and market-based food systems. At the same time, four Indian food systems have an issue of overconsumption. Protein and fat intake follows the recommendation in production- and subsidy-based food systems. For most of the market-based food systems, protein and fat intake exceeds recommendations. Home produced and subsidized food is negatively associated with dietary diversity in India. In comparison, market-purchased food and income are associated positively with dietary diversity. Overall, the environmental footprints of Indian food systems are proportionate to the calorie intake. Adopting healthy diets and sustainable food production is essential for fixing the broken food systems. Further studies on the transition of Indian food systems over time are necessary for addressing malnutrition and environmental sustainability in India.

## Supporting information

**S1 Appendix.**
(PDF)

## Author Contributions

**Conceptualization:** Tushar Ramchandra Athare, Prajal Pradhan, S. R. K. Singh, Juergen P. Kropp.

**Formal analysis:** Tushar Ramchandra Athare.

**Funding acquisition:** Tushar Ramchandra Athare.

**Investigation:** Tushar Ramchandra Athare.

**Methodology:** Tushar Ramchandra Athare, Prajal Pradhan, Juergen P. Kropp.

**Resources:** Juergen P. Kropp.

**Supervision:** Prajal Pradhan, Juergen P. Kropp.

**Validation:** Prajal Pradhan, S. R. K. Singh, Juergen P. Kropp.

**Visualization:** Prajal Pradhan.

**Writing – original draft:** Tushar Ramchandra Athare.

**Writing – review & editing:** Prajal Pradhan, S. R. K. Singh, Juergen P. Kropp.

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
