## [Decision Letter · Decision Letter 0]

28 Sep 2021

PONE-D-21-24663India consists of production-, subsidy-,  and market-based food systems with scoioeconomic and environmental variationsPLOS ONE

Dear Dr. Athare,

Thank you for submitting your manuscript to PLOS ONE. After careful consideration, we feel that it has merit but does not fully meet PLOS ONE’s publication criteria as it currently stands. Therefore, we invite you to submit a revised version of the manuscript that addresses the points raised during the review process.

The manuscript is very interesting. However, there is further need of improvement in presentation of results with regional/state level findings. There is also a need of strengthening the conclusion and policy recommendations.

We look forward to receiving your revised manuscript.

Kind regards,

Abid Hussain

Academic Editor

PLOS ONE

Journal Requirements:

T. R. Athare’s PhD is funded by Netaji Subhas-ICAR International Fellowship of Indian Council of Agricultural Research (ICAR), New Delhi. P. Pradhan acknowledges funding from the German Federal Ministry of Education and Research for the BIOCLIMAPATHS project (grant agreement No 01LS1906A) under the Axis-ERANET call and the German Federal Ministry for the Environment, Nature Conservation, Building, and Nuclear Safety for the I-CCC project (Contract No 81227263). The funders had no role in study design, data analysis, and preparation of the manuscript

T. R. Athare's PhD is funded by Netaji Subhas-ICAR International Fellowship of Indian Council of Agricultural Research (ICAR), New Delhi. 

P.\\ Pradhan acknowledges funding from the German Federal Ministry of Education and Research for the BIOCLIMAPATHS project (grant agreement No 01LS1906A) under the Axis-ERANET call and the German Federal Ministry for the Environment, Nature Conservation, Building, and Nuclear Safety for the I-CCC project (Contract No 81227263). 

The funders had no role in study design, data analysis, and preparation of the manuscript.

Reviewers' comments:

Reviewer's Responses to Questions

**Comments to the Author**

1. Is the manuscript technically sound, and do the data support the conclusions?

Reviewer #1: Yes

Reviewer #2: Yes

2. Has the statistical analysis been performed appropriately and rigorously? 

Reviewer #1: Yes

Reviewer #2: Yes

3. Have the authors made all data underlying the findings in their manuscript fully available?

Reviewer #1: Yes

Reviewer #2: Yes

4. Is the manuscript presented in an intelligible fashion and written in standard English?

Reviewer #1: Yes

Reviewer #2: Yes

5. Review Comments to the Author

Reviewer #1: It is a good piece of work covering a relatively less explored research topic of the "food system" in the Indian context.

Some minor comments need to be addressed :

1.Consider rewriting the title, it's not conveying the gist of the article with clarity.

2.The spatial dimensions in this manuscript is missing; authors have analyzed 0.2 Million samples spread across ~8k villages. However, only summarized results are shown; not providing any regional details - not even state wise. I suggest doing some efforts to present results spatially.

3.Authors can try choropleth mapping (Production, Subsidy and Market) to show food system dominance by the district.

4.Figure 1 (and fig. 3, fig. 5): Worth elaborating or finding the reasons why SIFPD not translating into SIDD especially for production-based food systems (PA), and subsidy-based as well as market-based food systems.

5.Similarly worth highlighting the reasons why water and GHG footprints vary significantly among the sub-systems? This would bring more clarity in targeting/promoting the food systems considering

6.Presentation style: The article uses single charting style - only bar graphs. Tables and maps are absent. Summarized results by the state would certainly add value to the article.

Reviewer #2: This study provides a holistic understanding of various food systems of Indian households, namely production, subsidy, market-based food systems. This paper is publishable after addressing the following issues.

The Introduction could relate to the broader discourse on undernutrition, overnutrition, and nutritional deficiencies in the various sectors of Indian society. This is particularly important in the context of increasing non-communicable lifestyle diseases, such as obesity, type 2 diabetes, certain cancers, hypertension, heart diseases, and mental health, that are related to diet and nutrition. The current introduction is very limited in scope to demonstrate the societal value of this research.

A question arises whether the survey reported in this paper is the latest available survey. It is also helpful to address the limitations of using existing surveys like this. It is not clear from the analysis if this survey also includes data on meat and fish consumption. This paper could benefit from an analysis of increasing meat, fish, and dairy consumption among the emerging middle-class consumers in India.

The analysis of calorie intake by households with the three food systems could have been more clearer. It is particularly important to explain why those with household production have lower dietary diversity and what policy interventions can make this food system responsive to household prosperity, human health, and the natural environment.

Last but not least, the authors may like to strengthen the conclusion and policy recommendations. Revision of the Introduction along the line suggested above will be helpful to rewrite the Conclusion.

6. PLOS authors have the option to publish the peer review history of their article (what does this mean?). If published, this will include your full peer review and any attached files.

Reviewer #1: No

Reviewer #2: No

---

## [Author Response · Author response to Decision Letter 0]

10 May 2022

We attach the comments to the reviewer in a separate file named response to reviewers and cover letter addressed to the editor.

---

## [Editor Report · Decision Letter 1]

9 Jun 2022

India consists of multiple food systems with scoioeconomic and environmental variations

PONE-D-21-24663R1

Dear Dr. Athare,

We’re pleased to inform you that your manuscript has been judged scientifically suitable for publication and will be formally accepted for publication once it meets all outstanding technical requirements.

Kind regards,

Abid Hussain

Academic Editor

PLOS ONE
---

## [Editor Report · Acceptance letter]

25 Jul 2022

PONE-D-21-24663R1 

India consists of multiple food systems with scoioeconomic
and environmental variations 

Dear Dr. Athare:

I'm pleased to inform you that your manuscript has been deemed suitable for publication in PLOS ONE. Congratulations! Your manuscript is now with our production department. 

Kind regards, 

on behalf of

Dr. Abid Hussain 

Academic Editor

PLOS ONE